# Probiotic bacteria vs. yeast for gastrointestinal diseases treatment: Protocol for a systematic review and meta-analysis

**Wardah Alsalemi**[1], **Zamri Chik**[1,2]*, **Mohammed Abdullah Alshawsh**[1,3]*

1 Department of Pharmacology, Faculty of Medicine, Universiti Malaya, Kuala Lumpur, Malaysia,
2 Universiti Malaya Research Centre for Biopharmaceuticals and Advanced Therapeutics (UBAT), Department of Pharmacology, Faculty of Medicine, Universiti Malaya, Kuala Lumpur, Malaysia,
3 Department of Paediatrics, Faculty of Medicine, Nursing and Health Sciences, Monash University, Monash Children's Hospital, Clayton, Victoria, Australia

* zamrichik@ummc.edu.my (ZC); mohammed.alshawsh@monash.edu, alshaweshmam@um.edu.my (MAA)

## Abstract

### Background

Disruption of the gut microbiota, an essential ecosystem of microorganisms inhabiting the gastrointestinal tract (GIT), has been linked to the development and progression of various gastrointestinal disorders. Probiotics have received considerable attention for their role in gastrointestinal diseases. However, there is a need to synthesise existing evidence to determine the optimal probiotic approach for managing complications of GIT disorders. By conducting a systematic review of randomised controlled trials (RCTs) comparing the effects of probiotic bacteria and yeast in patients with gastrointestinal diseases, we aim to provide a comprehensive and evidence-based analysis of the benefits and limitations of these interventions, which could inform clinical decision-making and improve patient outcomes in this population.

### Methods

Different databases, including PubMed, Web of Science, and Scopus will be searched to identify randomized controlled trials (RCTs). The titles and abstracts will be screened using Rayyan, and data will be extracted from eligible studies using Microsoft Excel. Critical appraisal and quality assessment will be performed using the ROB 2 tool, while GradePro will be used to assess the certainty of outcomes. All steps will be independently performed by two reviewers. This study will assess the effectiveness of yeast and bacterial probiotics in treating GIT disorders by evaluating inflammation markers, diarrhoeal score and disease severity, stool frequency, length of hospital stay, and adverse effects. By comparing the efficacy of probiotic bacteria and yeast, this review will identify the most effective type of probiotics for different

**Data availability statement:** No datasets were generated or analyzed during the current study. All relevant data from this study will be made available upon study completion.

**Funding:** This study was funded by the Malaysian Ministry of Higher Education through the Fundamental Research Grant Scheme (FRGS) (Grant Number: FRGS/1/2019/SKK10/UM/02/3). The grant was awarded to Prof Mohammed Abdullah Alshawsh. The funder's website is available at https://www.mohe.gov.my. The funders had no role in study design, data collection and analysis, decision to publish, or preparation of the manuscript.

**Competing interests:** The authors have declared that no competing interests exist.

**Abbreviations:** RCTs: randomized controlled trials; GIT: gastrointestinal tract; IBS: irritable bowel syndrome; IBD: inflammatory bowel disease; AAD: antibiotic-associated diarrhoea; WHO: World Health Organization; CI: confidence intervals; GRADE: grading of recommendations assessment development and evaluation; PRISMA-P: preferred reporting items for systematic reviews and meta-analyses protocols; PROSPERO: International Prospective Register of Systematic Reviews.

gastrointestinal disorders, potentially enhancing treatment outcomes and reducing healthcare costs.

**Systematic review registration**: PROSPERO (CRD42023384070).

## 1. Introduction

Gastrointestinal disorders, such as irritable bowel syndrome (IBS), inflammatory bowel disease (IBD), and antibiotic-associated diarrhoea (AAD), affect millions of people worldwide [1]. Between 1990 and 2019, there were 7.32 billion incidents and 2.86 billion prevalent cases of digestive diseases worldwide, resulting in 8 million deaths and 277 million disability-adjusted life years lost [2]. Gastrointestinal disorders are common health conditions that can arise due to various factors, including diet, stress, and certain medications [3]. A variety of studies have demonstrated the beneficial effects of probiotics on several gastrointestinal ailments, including diarrhoea, constipation, and IBS [4].

The gastrointestinal tract contains the most abundant and diverse microbial community in the human body. Over the past few decades, the interaction between gut microbiota, nutrition, and its impact on human health has garnered increasing interest [5]. Gut dysbiosis has been linked to the development or progression of many gastrointestinal conditions, such as IBD, ulcerative colitis, chronic inflammatory bowel disease, traveller's diarrhoea, and antibiotic-associated diarrhoea [6–10]. Undoubtedly, diarrhoea is one of the most common causes of intestinal dysfunction and can lead to dehydration and poor nutrition [11]. Constipation is another issue that can cause significant discomfort and disrupt regular daily activities [12]. IBS is a functional intestinal disorder characterised by abdominal pain, bloating, and changes in bowel movements [13]. These conditions can have a negative impact on an individual's quality of life and may result in the consumption of prescription medications for symptom relief. However, research suggests that using probiotics may be beneficial for alleviating the symptoms of gastrointestinal disorders and improving the quality of life of individuals with these conditions [14,15].

The microbiota plays important roles in human health, including human behavioural responses, homeostasis, digestion of food, and shielding of the human gut against pathogens. In healthy adults, there are two main phyla of endogenous bacteria, Bacteroidetes and Firmicutes, which account for approximately 90% of the gut microbiome [16]. Probiotics are defined as "live microorganisms, which when administered in adequate amounts, confer a health benefit on the host" [17]. In other words, they are non-pathogenic microorganisms that positively influence the health or physiology of the host when consumed. They consist of either yeast or bacteria, mainly lactic acid bacteria [18].

Probiotic bacteria such as *Lactobacillus* and *Bifidobacterium* are non-pathogenic lactic acid bacteria that have been an integral component of the human diet for centuries [19]. The consumption of these probiotics enhances the diversity and richness of the microbiome and reduces the presence of pathogenic bacteria [20]. A favourable alteration in the composition of the intestinal microbiota is directly correlated with a decreased risk of developing colorectal cancer. *Lactobacillus* and *Bifidobacterium* species have been extensively studied for their potential to improve gut health, boost

immune function, and alleviate the symptoms associated with gastrointestinal disorders [21]. In addition, probiotics can reduce inflammation in the gut, which is a critical component of IBD. Bacterial probiotics have also been shown to prevent and treat antibiotic-associated diarrhoea, which is a common side effect of antibiotics [22]. These probiotics prevent the colonisation of harmful bacteria in the gut and help restore the balance of beneficial bacteria. The beneficial properties of probiotic bacteria vary based on their strain, dosage, ability to survive in the gut, and mechanism of action [23]. It has been reported that using a multi-strain mixture of probiotics is more effective than using a single strain, as this allows bacterial strains to work synergistically [24]. Several multi-strain probiotics are commercially available and mainly contain strains from the *Lactobacillus* and *Bifidobacterium* genera.

Similarly, probiotic yeasts, such as *Saccharomyces boulardii*, have shown numerous beneficial effects on metabolism, human physiology and immune homeostasis in the colon and possess anti-inflammatory, antiproliferative, and anticancer properties [25]. *Saccharomyces boulardii* has been prescribed for the past 30 years to prevent and treat bacterial diarrhoeal diseases. Notably, *S. boulardii* has demonstrated clinical and experimental effectiveness in gastrointestinal disorders, with a predominant inflammatory component, indicating that this probiotic might interfere with cellular signalling pathways common to various inflammatory conditions [26]. Several mechanisms of action have been identified targeting both the host and pathogenic microorganisms. They include the regulation of intestinal microbial homeostasis, interference with the ability of pathogens to colonise and infect the mucosa, modulation of local and systemic immune responses, stabilisation of gastrointestinal barrier function, and induction of enzymatic activity that favours absorption and nutrition. Several clinical trials and experimental studies strongly suggest the use of *Saccharomyces boulardii* as a biotherapeutic agent for the prevention and treatment of several gastrointestinal diseases [14,27,28].

While bacterial probiotics are widely recognized as supplements for gastrointestinal health, yeast probiotics such as *Saccharomyces boulardii* have also shown promising therapeutic benefits [29]. To evaluate the efficacy and safety of both yeast and bacterial probiotics in treating gastrointestinal disorders, conducting a systematic review of RCTs is crucial to provide a comprehensive and evidence-based analysis of their benefits and limitations. Randomized controlled trials (RCTs) are widely regarded as the gold standard for evaluating clinical interventions due to their ability to minimize bias through random allocation. By ensuring that treatment and comparator groups are similar in all respects other than the intervention, RCTs enhance internal validity and causal inference. However, RCTs may also have limitations, particularly in subgroups analyses based on factors such as age, gender, or ethnicity, where small sample sizes may reduce statistical power and generalizability. In this protocol, we acknowledge these limitations and will conduct subgroup analyses and meta-regression, where sufficient data are available, to explore differential effects and reduce potential selection bias.

Despite numerous individual studies on probiotics for gastrointestinal diseases, there remains a gap in synthesizing the most effective strains and dosages, which this review aims to address. This systematic review will synthesize the available evidence and critically appraise the quality of the studies to ensure that the conclusions are based on robust and unbiased assessments. Understanding the optimal type and dosage of probiotics for specific gastrointestinal disorders is essential. The findings could significantly impact clinical practice and public health, offering valuable guidance for both clinicians and patients. This protocol outlines the methodology for a systematic review evaluating the efficacy of bacterial and yeast probiotics in improving outcomes for patients with gastrointestinal disorders, focusing on parameters, such as inflammation markers, diarrheal scores, gut microbiota modulation and disease severity.

## 2. Methods

### 2.1. Protocol registration

The protocol for this systematic review adheres to the Preferred Reporting Items for Systematic Reviews and Meta-Analysis Protocols (PRISMA-P) guidelines [30] (Fig 1 and S1 Table). Furthermore, it was registered with the "*International Prospective Register of Systematic Reviews*" (PROSPERO) under registration number (CRD42023384070).

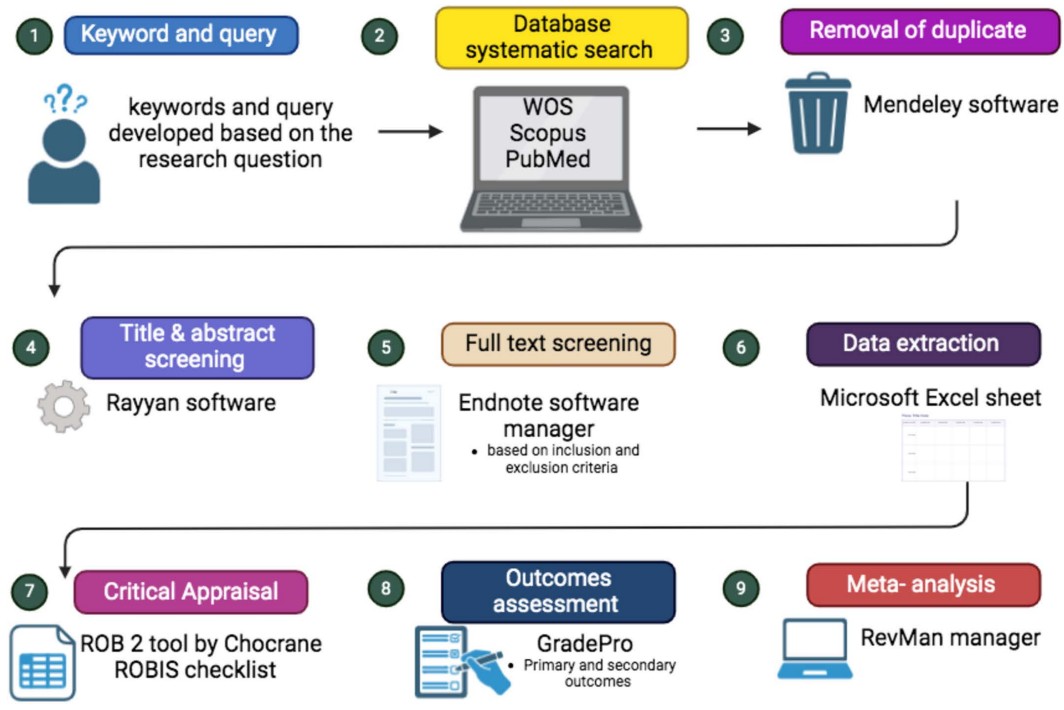

**Fig 1. Illustration of the steps of the systematic review protocol (created with BioRender.com).**

### 2.2. Search strategies

Relevant studies will be identified using search terms and Medical Subject Headings (MeSH) tailored to each database, based on the most recent literature review . RCTs will be identified from PubMed, Scopus, and Web of Science, and the search results will be filtered to include only RCTs. All steps will be independently performed by two reviewers. Literature searches will be meticulously conducted using a combination of search terms and MeSH connected by Boolean AND/OR operators as follows:

1    # (yeast OR acidophilus OR animalis OR Bacillus OR Bifidobacteri* OR boulardii OR breve OR bulgaricus OR casei OR cerevisiae OR Clostridium OR Enterococcus OR Escherichia OR faecalis OR helveticus OR infantis OR Lactobacill* OR longum OR paracasei OR plantarum OR Probiotic* OR Propionibacteri* OR rhamnosus OR saccharomyces OR salivarius OR Streptococcus* OR synobiotic OR thermophilus OR "Rhodosporidium paludigenum" OR Yakult OR "fermented milk")

2    # (formula OR supplementation OR strain* OR mixture OR capsule OR sachet OR powder OR food OR liquid)

3    # ("gastrointestinal disorder*" OR "gastrointestinal disease*" OR "functional gastrointestinal disorder" OR fgid OR "functional bowel" OR "colonic disease" OR "irritable colon" OR "irritable bowel syndrome" OR IBS OR "inflammatory bowel diseases" OR "ulcerative colitis" OR "Crohn's disease" OR diarrhea OR diarrhoea OR "traveller diarrhea" OR "antibiotic-associated diarrhea" OR "mucous colitis" OR "lactose intolerance" OR "necrotising enterocolitis" OR Dysbiosis OR "spastic colon" OR "gastro-esophage* reflux" OR "Helicobacter pylori" OR "H. pylori" OR GERD OR "antibiotic resistance" OR "colon cancer" OR "rectal cancer" OR "colon carcinoma" OR "rectal carcinoma" OR "colorectal cancer" OR "colorectal carcinoma" OR "colonic cancer" OR "colonic carcinoma")

4  # ("Clinical trial*" OR trial* OR "intervention study" OR RCT OR "randomi* control* trial" OR "randomi* clinical trial" OR "double blinded" OR placebo)

5  # (comparative OR comparison* OR comparing OR compared OR compare OR "compared with" OR versus OR Vs OR against)

## 2.3. Inclusion and exclusion criteria

**2.3.1. Participants.** This review includes human participants across all age groups, genders, and ethnic backgrounds who have been diagnosed with a gastrointestinal disorder. Specific conditions considered include, but are not limited to, *Helicobacter pylori* infection, Crohn's disease, IBS, IBD, ADD, Traveller's diarrhoea and ulcerative colitis (Table 1). Participants must have received probiotic supplementation as part of their therapeutic regimen targeting their gastrointestinal disorder. The exclusive focus on human participants ensures that findings are directly applicable to clinical and real-world settings, enhancing the relevance to patient care and treatment outcomes. Studies involving non-human subjects, such

**Table 1. The inclusion and exclusion criteria based on PICOs model.**

| PICOs acronym | Inclusion | Exclusion |
|---|---|---|
| Population or participants | • Human only, with any age, ethnicity, and gender<br>• Diagnosed with any type of gastrointestinal disorder such as (*Helicobacter pylori* infection,Diarrhoea, Ulcerative colitis, etc.)<br>• Received a probiotic supplementation. | • Animal models<br>• Cell lines<br>• Patients received probiotics for other diseases not related to gastrointestinal disorders. |
| Intervention | • Any bacterial probiotic species, e.g.,; (Lactobacillus acidophilus, Bifidobacterium longum, Propionibacterium, Bacillus clausii, Bacillus coagulans, Bifidobacterium breve, Bifidobacterium lactis, Lactobacillus paracasei, Lactobacillus reuteri, Lactobacillus rhamnosus)<br>• Taken as a supplement (capsule, sachet, powder)<br>• Used on its own or in combination such as (Visibiome, Vivomixx, VSL, Yakult). | • Antibiotic treatment, inulin, kombucha, butylated starch, fermented milk consumption, guar gum supplementation, human milk oligosaccharide, oligofructose-enriched inulin, selenium-derived yeast, and Wheat bran extract (WBE)<br>• Probiotic formula not compared with yeast formula.<br>• Probiotics combined with other types of treatment.<br>• Synobiotic fermented milk<br>• Prebiotics |
| Comparator | • Comparing bacterial probiotics and *Saccharomyces cerevisiae* OR *Saccharomyces boulardii* or any other probiotic formula containing yeast. | • Placebo<br>• Untreated group |
| Outcomes | **Primary outcomes:**<br>• Diarrheal score – Bristol scoring system.<br>• Disease progression/Recovery period<br>• Gut microbiota modulation/ diversity (16s RNA sequencing)<br>• Pro and/or anti-Inflammatory markers<br>• Severity of the disease (assessed with the IBS Severity Scoring System (IBS-SSS)<br>• Stool consistency and/or frequency – using stool charts, diaries, collections, and recall.<br>Secondary outcomes:<br>• Mortality rate – deaths associated with gastrointestinal infection or inflammation.<br>• Adverse effects<br>• Quality of life | |
| Study designs | • A randomized controlled trial<br>• Randomized clinical trial | • All other types of study (e.g.,; observational, cohort, case-control, etc.) |
| Additional criteria | • Last 10 years (up to 2025)<br>• Only original articles (primary studies) | • All secondary studies (Systematic review, narrative review, any type of review)<br>• Conferences, thesis, books, chapters. |

as animal models or in *vitro* cell lines, are excluded to maintain the applicability of results to human physiology and health outcomes. Additionally, studies in which probiotics were administered for non-gastrointestinal conditions (e.g., respiratory or urinary tract infections or neurological disorders) are excluded, preserving a targeted focus on gastrointestinal disorders.

**2.3.2. Intervention.** The intervention of interest is probiotic supplementation comprising bacterial species. This includes well-documented strains but not limited to such as *Lactobacillus acidophilus, Bifidobacterium longum, Propionibacterium, Bacillus clausii, Bacillus coagulans, Bifidobacterium breve, Bifidobacterium lactis, Lactobacillus paracasei, Lactobacillus reuteri, and Lactobacillus rhamnosus.* There is no restriction in dose and formulation, probiotics may be administered in forms such as capsules, sachets, powders, or liquids in different doses. Single-strain and multi-strain formulations, including commercial products (e.g., Visbiome®, Vivomixx®, VSL#3®, AgiMixx® Culturelle® Bio-K+° and Yakult®) are eligible, allowing for a comprehensive evaluation of bacterial probiotics' impact on gastrointestinal health. Excluded interventions encompass treatments involving probiotics combined with conventional treatment or antibiotics. Also, probiotic containing other components like prebiotics such as inulin, fermented beverages (e.g., kombucha), or any non-bacterial supplements (e.g., guar gum, human milk oligosaccharides, or wheat bran extract) will be excluded. This will ensure a focused assessment of bacterial probiotics specifically.

**2.3.3. Comparator.** Studies that compare bacterial probiotics to yeast-based probiotics, such as *Saccharomyces cerevisiae or Saccharomyces boulardii*, are included. This approach facilitates an analysis of efficacy differences between bacterial and yeast probiotics in managing gastrointestinal disorders. Studies comparing probiotics with a placebo or untreated group are excluded, as this systematic review protocol targets the comparison of bacterial versus yeast probiotics, rather than evaluating probiotics against a placebo.

**2.3.4. Outcomes.**

**I. Primary Outcomes:**

- **Diarrheal Score:** Assessed via standardized scoring tools such as the Bristol Stool Chart [31].

- **Disease Progression and Recovery Period:** Evaluates how probiotics influence symptom reduction and recovery speed [32].

- **Gut Microbiota Modulation/Diversity:** Changes in microbiota diversity, typically measured through 16S rRNA sequencing [33].

- **Pro- and Anti-Inflammatory Markers:** Cytokine profiles and C-reactive protein levels to measure inflammatory modulation. Pro-Inflammatory Markers, e.g., (Tumor Necrosis Factor-alpha (TNF-α), Interleukin-1 beta (IL-1β), Interleukin-6 (IL-6), Interleukin-8 (IL-8), Interferon-gamma (IFN-γ), Nuclear Factor-kappa B (NF-κB) Anti-Inflammatory Markers, e.g., (Interleukin-10 (IL-10), Transforming Growth Factor-beta (TGF-β), Interleukin-1 Receptor Antagonist (IL-1Ra) [34].

- **Disease Severity:** Severity scales such as the IBS Severity Scoring System (IBS-SSS) [35].

- **Stool Consistency and Frequency:** Patient-centered gastrointestinal symptom tracking via stool diaries [36].

**II. Secondary Outcomes:**

- **Mortality Rate:** Assessing the impact of probiotics on mortality related to gastrointestinal disease [37].

- **Adverse Effects:** Probiotic safety and tolerability based on recorded side effects [38].

- **Quality of Life:** Patient-reported outcomes on overall well-being and functional improvements [39].

**2.3.5. Study design.** The review includes randomized controlled trials (RCTs), which are chosen for their robust methodologies to minimize bias and reliably assess probiotics' causal effects on gastrointestinal disorders. Observational

studies, cohort studies, and case-control studies are excluded as they lack the randomization required to establish causation, thereby focusing on high-quality evidence from randomized trials. Only studies published within the last 10 years (up to 2025) are considered, ensuring relevance to current clinical practices. Furthermore, only primary research articles are included to prioritize first-hand data. Secondary research, including systematic and narrative reviews, as well as conference proceedings, theses, and book chapters, are excluded due to potential peer-review and methodological limitations.

## 2.4. Selection process of the included studies

The titles and abstracts of the primary studies obtained from the databases will be screened based on the eligibility criteria using the AI-powered Rayyan platform (https://www.rayyan.ai) that is designed to streamline systematic literature reviews by identifying relevant studies quickly since it reduces screening time up to 90%. After removing duplicate articles, studies will be classified into three categories: relevant, irrelevant, or uncertain. Studies will be marked as "uncertain" if the title and abstract lack sufficient information to determine eligibility, such as missing details about the type of probiotic used, comparator group, or targeted gastrointestinal condition. These studies will undergo full-text review for a final inclusion decision.

Studies classified as irrelevant by two reviewers will be excluded. The full texts of the remaining studies will be reviewed independently by two reviewers using Endnote software X9 (https://endnote.com/), with inclusion based on adherence to eligibility criteria. In cases of discrepancies, reviewers will first discuss to reach a consensus, with a third opinion sought if needed. The numbers of included and excluded studies along with reasons for exclusions will be reported using the PRISMA flow chart.

## 2.5. Data extraction

The data will be extracted and consistently recorded using a standardized data extraction sheet. The most important data will be organized into a table of study characteristics and outcomes. Data extraction will be performed independently by two authors. The following data will be extracted from each study: title, DOI number, authors, year, study characteristics (date of study recruitment, study design, number of arms, method of patient randomization, patient selection, exclusion, and inclusion criteria), participant characteristics (age, sex, sample size, type of disorder and complications, and number of withdrawals with/without reasons), intervention details (number of groups, probiotic specifications, formula, strains, dose, duration, frequency, and, where available, demographic-specific dosing patterns such as by age or gender), and outcomes (primary and secondary outcomes, parameters measured, samples collected, follow-up, findings with SD or SEM and $P$ value for each outcome, and adverse events).

## 2.6. Risk of bias assessment

The revised Cochrane risk of bias tool for randomized trials (RoB-2) will be used to determine the risk of bias in the included studies [40]. The risk of bias will be evaluated independently by two reviewers by considering the following domains: (1) bias arising from the randomization process (selection bias); (2) bias due to deviations from intended interventions (performance bias); (3) bias due to missing outcome data (attrition bias); (4) bias in measurement of the outcome (detection bias); (5) bias in selection of the reported result (reporting bias); and overall bias.

## 2.7. Quality of evidence

The quality of evidence for each individual outcome reported in the included studies will be assessed according to the Grading of Recommendation Assessment, Development, and Evaluations (GRADE) guidelines using GRADE Pro (www.gradepro.org) [41]. The GRADE system categorizes the quality of evidence into four levels: high, moderate, low, and very

low. GRADE considers factors, including the risk of bias, indirectness, inconsistency, imprecision, and publication bias. Two researchers will independently perform the quality assessment and GRADE evaluation. Any disagreements will be resolved through discussion, with the involvement of a third researcher if necessary.

## 2.8. Strategy for data synthesis and analysis

If the included studies demonstrate sufficient homogeneity, the data will be pooled, and a quantitative synthesis (meta-analysis) will be conducted. The meta-analysis will be conducted using Review Manager (RevMan, Version 5.3, as recommended by the Cochrane Handbook). [42]. Continuous data outcome measures will be compared between the intervention and comparator groups (bacterial vs. yeast probiotics) using a random effects model. The standardized mean difference with 95% confidence intervals (CIs) will be used to analyse intervention effects. For dichotomous outcomes, the relative risk with 95% CI will be utilized to analyse treatment effects. Heterogeneity will be assessed using $I^2$; an $I^2$ value of less than 30% indicates low heterogeneity, 30% to 65% suggests possible moderate heterogeneity, and > 65% indicates substantial heterogeneity. If substantial heterogeneity is observed ($I^2 > 65\%$), subgroup analyses will be conducted to identify potential sources of heterogeneity, such as probiotic strain, dosage, intervention duration, and disease type. Where data are sufficient, subgroup analysis or meta-regression will also examine the influence of dosage across specific demographic variables such as (e.g., age, gender) to assess potential patterns or differential effects. To account for possible subgroup differences (e.g., age, gender, ethnicity) that may lead to imbalances in some included trials, subgroup analyses and meta-regression will be conducted where sufficient data exist. These analyses will help explore sources of heterogeneity and potential selection bias.

Sensitivity analysis will be carried out to assess the source of heterogeneity, including the risk of bias, fixed model vs. random model, types of probiotics, and duration of intervention. Comparisons between specific probiotic strains (e.g., *Lactobacillus* vs. *Saccharomyces*) will be considered exploratory and hypothesis-generating. Additionally, sensitivity analyses will be conducted to assess the robustness of findings by examining the impact of removing RCTs (leave-one-out analyses) with small sample size. Since a meta-analysis with a small number of studies may still yield meaningful results if effect size is large and confidence intervals are precise, small-study effects and publication bias will be considered if fewer than 10 RCTs are included in this systematic review.

### 2.8.1. Meta-analysis procedure using review manager.
The meta-analysis will be conducted using Review Manager (RevMan, Version 5.3), developed by the Cochrane Collaboration. This tool facilitates the synthesis of quantitative data from multiple studies and supports rigorous statistical analysis. The process begins with data entry, where the mean difference or effect sizes and corresponding standard deviations or confidence intervals from the included RCTs will be entered into RevMan for each outcome of interest. Next, a random-effects model will be selected to account for between-study variability and for effect estimates, continuous outcomes will be analyzed using standardized mean differences (SMD) with 95% confidence intervals, while dichotomous outcomes will be analyzed using relative risk (RR) with 95% confidence intervals.

Heterogeneity will be assessed using the $I^2$ statistic, with thresholds defined as <30% (low), 30–65% (moderate), and >65% (substantial). To assess publication bias, funnel plots will be generated in RevMan, and Egger's regression test will be used to detect any potential bias in the included studies.

## 2.9. Dealing with missing data

If the included study fails to include all required data as per the data extraction form, efforts will be made to contact the first author or corresponding author via email to request the missing data. If author contact is unsuccessful after multiple attempts, alternative strategies will be employed, including conducting sensitivity analyses by excluding studies with missing data and comparing results with and without them. We will also use available summary statistics (e.g., estimating standard deviations (SD) from reported SEM, confidence intervals, interquartile ranges, or *p*-values if provided) to handle

missing data. If missing data are more prevalent in certain subgroups (e.g., smaller studies or specific interventions), meta-regression will be used to assess whether missing data influence effect estimates. Finally, if key studies lack essential data and cannot be included quantitatively, their findings will be synthesized narratively rather than through quantitative analysis.

### 2.10. Publication bias

Publication bias will be assessed using a combination of funnel plots, which will be generated utilizing Review Manager 5.3 software, and Egger's regression test [43].

### Supporting information

**S1 Table. PRISMA-P 2015 Checklist.**
(PDF)

### Author contributions

**Conceptualization:** Mohammed Abdullah Alshawsh.

**Data curation:** Wardah Alsalemi.

**Formal analysis:** Wardah Alsalemi.

**Funding acquisition:** Zamri chik, Mohammed Abdullah Alshawsh.

**Investigation:** Zamri chik, Mohammed Abdullah Alshawsh.

**Methodology:** Wardah Alsalemi.

**Project administration:** Zamri chik, Mohammed Abdullah Alshawsh.

**Resources:** Zamri chik, Mohammed Abdullah Alshawsh.

**Supervision:** Zamri chik, Mohammed Abdullah Alshawsh.

**Validation:** Mohammed Abdullah Alshawsh.

**Visualization:** Wardah Alsalemi.

**Writing – original draft:** Wardah Alsalemi.

**Writing – review & editing:** Wardah Alsalemi, Zamri chik, Mohammed Abdullah Alshawsh.

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
