## [Decision Letter · Decision Letter 0]

24 Feb 2025

PONE-D-24-50095Probiotic Bacteria vs. Yeast for Gastrointestinal Diseases Treatment: Protocol for a Systematic Review and Meta-AnalysisPLOS ONE

Dear Dr. Alshawsh,

Thank you for submitting your manuscript to PLOS ONE. After careful consideration, we feel that it has merit but does not fully meet PLOS ONE’s publication criteria as it currently stands. Therefore, we invite you to submit a revised version of the manuscript that addresses the points raised during the review process.

We look forward to receiving your revised manuscript.

Kind regards,

Awatif Abid Al-Judaibi, PhD

Academic Editor

PLOS ONE

Journal requirements: 1. When submitting your revision, we need you to address these additional requirements. Please ensure that your manuscript meets PLOS ONE's style requirements, including those for file naming. The PLOS ONE style templates can be found at https://journals.plos.org/plosone/s/file?id=wjVg/PLOSOne_formatting_sample_main_body.pdf and https://journals.plos.org/plosone/s/file?id=ba62/PLOSOne_formatting_sample_title_authors_affiliations.pdf

Reviewers' comments:

Reviewer's Responses to Questions

**Comments to the Author**

1. Does the manuscript provide a valid rationale for the proposed study, with clearly identified and justified research questions?

Reviewer #1: Partly

Reviewer #2: Yes

Reviewer #3: Yes

2. Is the protocol technically sound and planned in a manner that will lead to a meaningful outcome and allow testing the stated hypotheses?

Reviewer #1: Yes

Reviewer #2: Yes

Reviewer #3: Partly

3. Is the methodology feasible and described in sufficient detail to allow the work to be replicable?

Reviewer #1: Yes

Reviewer #2: Yes

Reviewer #3: Yes

4. Have the authors described where all data underlying the findings will be made available when the study is complete?

Reviewer #1: Yes

Reviewer #2: Yes

Reviewer #3: No

5. Is the manuscript presented in an intelligible fashion and written in standard English?

Reviewer #1: Yes

Reviewer #2: Yes

Reviewer #3: Yes

6. Review Comments to the Author

You may also provide optional suggestions and comments to authors that they might find helpful in planning their study.

**Reviewer #1: ** Dear Authors,

When I reviewed your article titled "Probiotic Bacteria vs. Yeast for Gastrointestinal Diseases Treatment: Protocol for a Systematic Review and Meta-Analysis" that you applied as a study protocol, I believe that it is a protocol that can be beneficial to the field of study, except for some minor corrections mostly related to the introduction part of your article.

You can find the corrections I suggested for your article below:

1- Citation is required for the information you provided in lines 38-41.

2- Your reference to the information you provided in lines 42-44 (citation number 3) is irrelevant. It should be cited with an appropriate study.

3- Your reference to the information you provided in lines 44-47 (citation number 4) is a quote from the introduction part of the reference article. The referenced article wrote this information by referencing other studies. It would be more accurate if you also referenced these studies. Likewise, when citing a reference, it would be more appropriate in terms of scientific work ethics to cite the method or results of the referred study.

4- Your reference to the information you provided in lines 58-60 (citation number 6) is a quote from the introduction of the reference article. My previous comment is also valid for this criticism.

5- Your reference to the information you provided in lines 60-62 (citation number 7) is irrelevant. It should be cited with an appropriate study.

6- Only one article was cited for the long and detailed information in lines 63-77. When the relevant reference was examined, it was seen that it did not fully cover the detailed information provided. The information provided should be supported with more appropriate articles by including the information in line 78 and cited.

7- The information provided in lines 93-34 is related to the fact that bacterial probiotics are more remarkable than probiotic yeasts as a supplement in gastrointestinal diseases. The relevant claim is a controversial issue, and citation no. 12 does not refer to this information. It should be cited with an appropriate article or the claim should be removed from the publication.

8- The sentence in lines 104-106 begins with "The purpose of this systematic review is...." Since this article is a study protocol article, the sentence should also start with the purpose of this study protocol. In addition, the rest of the sentence should be revised accordingly.

**Reviewer #2:**  Reviewer comment

1. Yes, the manuscript provides a valid rationale for the proposed study with clearly identified and justified research questions. The introduction outlines the academic problem gastrointestinal disorders and the role of probiotics in their management while emphasizing the need for a systematic review to compare bacterial and yeast probiotics. The research question is well-defined, focusing on the comparative effectiveness of these probiotic types, and is justified by the gaps in existing literature. The methodology is good, employing randomized controlled trials and critical appraisal tools to ensure reliability.

2. Yes, But still it needs some clarification because the protocol does not explicitly mention whether a formal power analysis has been conducted to determine the required sample size for meaningful statistical comparisons. While meta-analyses often rely on available studies, an estimation of expected effect sizes and required study numbers would strengthen the methodological justification. Additionally, although heterogeneity is acknowledged using the I² test, further details on managing substantial heterogeneity (>65%), such as subgroup analyses or meta-regression, would be beneficial. Lastly, the manuscript does not clearly indicate which analyses, if any, will be exploratory. Clarifying whether certain comparisons, such as probiotic strain-specific effects, are secondary or exploratory would enhance transparency.

3. Yes, however it is important addressing a few areas. First, it does not mention whether sample size estimation or power analysis has been performed, which is essential for ensuring the reproducibility and robustness of the study. Including a discussion on expected effect sizes or the number of studies needed would enhance the overall clarity and rigor. Additionally, while the protocol suggests contacting authors to handle missing data, it would be useful to specify alternative methods, such as imputation strategies, in case responses are not obtained.

4. Yes, but the authors could specify where the final dataset will be deposited. This would strengthen compliance with open data requirements.

5. Yes, the manuscript is intelligible and written in Standard English, but minor grammatical and stylistic improvements would enhance clarity and readability. It would benefit from professional proofreading or language editing before submission.

**Reviewer #3:**  Review report on “Probiotic Bacteria vs. Yeast for Gastrointestinal Diseases Treatment: Protocol for a Systematic Review and Meta-Analysis”

Manuscript ID: PONE-D-24-50095

Alsalem et al. have designed a systematic review protocol using randomized controlled trials (RCTs) to compare the effects of probiotic bacteria and yeast in patients with gastrointestinal diseases to identify the most effective type of probiotics for different gastrointestinal disorders, that may potentially enhance treatment outcomes with reduced healthcare costs. Regarding the protocol, I have a few concerns, which I am mentioning below:

1. The authors chose to apply “randomized clinical trials (RCTs)” in their study. They should at least define this approach in the background or upfront in the paper (not in the study design). They should also discuss about its pros and cons in brief.

In this context, we already know that in RCTs, the two or more groups of people in each trial should be as similar as possible, except for the received treatments, to ensure that any difference in the outcomes between the groups are solely due to the treatments they received. Since the authors aim to consider participants of any age or any gender or any ethnicity in these trials, I suspect that they will end up with extremely small (n = 2 to 10) sample size in only one group (received treatment group) for a specific age or gender in a few cases. Do they think that applying RCTs will help in drawing a fair unbiased conclusion in those cases? In other words, how will they deal with selection bias in that case? Which statistics they propose to apply for that comparative analysis?

2. I understand that the authors do not intend to apply any statistics to deal with the missing data, since it may introduce biases itself. However, the authors’ original plan to contact the first authors of the published papers for this systematic review protocol, seems to be an incompetent approach to deal with the missing data, especially when the publications are older (for ex., 4 to 10 years old publications). This solution may work if they consider mostly recent publications. What will be the impact of publication bias in that case? Otherwise, how do the authors plan to avoid attrition bias?

3. “After the removal of duplicate articles, studies will be classified into three categories: relevant, irrelevant, or uncertain” – which criteria help to identify uncertain articles? Please clarify.

4. “there remains a gap in synthesizing the most effective strains and dosages, which this review aims to address” – I understand that the designed protocol may help to identify the best type of probiotics for different gastrointestinal disorders. How does this protocol help to select appropriate dosage of probiotics with the available sample sizes for a specific gender or age?

5. How does Review Manager help in the meta-analyses? Please clarify this section step by step in the manuscript for the general readers.

Minor concerns:

1. Authors should briefly explain different tools or scores for the ease of general readers. For e.g., instead of mentioning only ‘Rayyan’ platform, the authors could write – “AI-powered ‘Rayyan’ platform, that is designed to streamline systematic literature reviews by identifying relevant studies quickly since it reduces screening time up to 90%”.

2. In line no. 243 of page no.10, the authors mention about ‘number of arms’. What does it mean?

3. How do the participant characteristics, i.e., the number of withdrawals with/without reasons (line 245 of page no. 10) help in this protocol?

4. The background lacks adequate references against most of the statements from previous findings. In many cases, they did not even provide a single reference (e.g., line no. 52 to 54 of page 3). For instance, when the authors mention “a variety of studies” (line no. 39 to 41 of page 2) or “several clinical trials and experimental studies” (line no. 90 to 92 of page 4), they should at least provide two or three references against their statements.

7. PLOS authors have the option to publish the peer review history of their article (what does this mean? ). If published, this will include your full peer review and any attached files.

**Do you want your identity to be public for this peer review?** For information about this choice, including consent withdrawal, please see our Privacy Policy .

Reviewer #1: No

Reviewer #2: No

Reviewer #3: No

---

## [Author Response · Author response to Decision Letter 1]

1 Apr 2025

The responses to the reviewers' and editor's comments are attached.

---

## [Decision Letter · Decision Letter 1]

20 Apr 2025

PONE-D-24-50095R1Probiotic Bacteria vs. Yeast for Gastrointestinal Diseases Treatment: Protocol for a Systematic Review and Meta-AnalysisPLOS ONE

Dear Dr. Mohammed Abdullah Alshawsh,

Thank you for submitting your manuscript to PLOS ONE. After careful consideration, we feel that it has merit but does not fully meet PLOS ONE’s publication criteria as it currently stands. Therefore, we invite you to submit a revised version of the manuscript that addresses the points raised during the review process.

We look forward to receiving your revised manuscript.

Kind regards,

Awatif Abid Al-Judaibi, PhD

Academic Editor

PLOS ONE

Journal Requirements:

Reviewers' comments:

Reviewer's Responses to Questions

**Comments to the Author**

1. Does the manuscript provide a valid rationale for the proposed study, with clearly identified and justified research questions?

Reviewer #1: Partly

Reviewer #2: Yes

Reviewer #3: Yes

2. Is the protocol technically sound and planned in a manner that will lead to a meaningful outcome and allow testing the stated hypotheses?

Reviewer #1: Yes

Reviewer #2: Yes

Reviewer #3: Yes

3. Is the methodology feasible and described in sufficient detail to allow the work to be replicable?

Reviewer #1: Yes

Reviewer #2: Yes

Reviewer #3: Yes

4. Have the authors described where all data underlying the findings will be made available when the study is complete?

Reviewer #1: Yes

Reviewer #2: Yes

Reviewer #3: Yes

5. Is the manuscript presented in an intelligible fashion and written in standard English?

Reviewer #1: Yes

Reviewer #2: Yes

Reviewer #3: Yes

6. Review Comments to the Author

You may also provide optional suggestions and comments to authors that they might find helpful in planning their study.

Reviewer #1: Dear Authors,

When I reviewed your revised article titled "Probiotic Bacteria vs. Yeast for Gastrointestinal Diseases Treatment: Protocol for a Systematic Review and Meta-Analysis" that you applied as a study protocol, again, I believe that it is a protocol that can be beneficial to the field of study. Except for my 3rd criticism (maybe you missed), I see the minor corrections that I suggested, mostly done. Therefore, I repeat my words: “The referenced article wrote this information by referencing other studies. It would be more accurate if you also referenced these studies. Likewise, when citing a reference, it would be more appropriate in terms of scientific work ethics to cite the method or results of the referred study.” I suggest one last minor correction that I suggested to you before. After that last correction, there won't be any additional criticism from me.

To remind my 3rd criticism (for the 1st version of your manuscript):

"3- Your reference to the information you provided in lines 44-47 (citation number 4) is a quote from the introduction part of the reference article. The referenced article wrote this information by referencing other studies. It would be more accurate if you also referenced these studies. Likewise, when citing a reference, it would be more appropriate in terms of scientific work ethics to cite the method or results of the referred study."

Reviewer #2: Thank you for the opportunity to review this protocol. The proposed study addresses a relevant and timely topic and has the potential to contribute meaningfully to the field.

Reviewer #3: I appreciate the authors for their efforts into revising the manuscript. Besides the major issues, the authors also worked on the minor issues. They have given proper justifications for the questions I asked and made changes at places in the manuscript to make it easier to follow. I am happy with the current version of the manuscript, and I thank the authors for their thorough revisions.

7. PLOS authors have the option to publish the peer review history of their article (what does this mean? ). If published, this will include your full peer review and any attached files.

**Do you want your identity to be public for this peer review?** For information about this choice, including consent withdrawal, please see our Privacy Policy .

Reviewer #1: No

Reviewer #2: **Yes: ** Addis Tekaw Birhenie

Reviewer #3: No

---

## [Author Response · Author response to Decision Letter 2]

29 Apr 2025

Thank you. We have replaced the citation and added the original studies referenced in that article. These sources are now correctly cited in the revised manuscript (Page 2

Line 50).

---

## [Decision Letter · Decision Letter 2]

4 May 2025

Probiotic Bacteria vs. Yeast for Gastrointestinal Diseases Treatment: Protocol for a Systematic Review and Meta-Analysis

PONE-D-24-50095R2

Dear Dr. Mohammed Abdullah Alshawsh,

We’re pleased to inform you that your manuscript has been judged scientifically suitable for publication and will be formally accepted for publication once it meets all outstanding technical requirements.

Kind regards,

Awatif Abid Al-Judaibi, PhD

Academic Editor

PLOS ONE

Reviewers' comments:

Reviewer's Responses to Questions

**Comments to the Author**

1. Does the manuscript provide a valid rationale for the proposed study, with clearly identified and justified research questions?

Reviewer #1: Partly

2. Is the protocol technically sound and planned in a manner that will lead to a meaningful outcome and allow testing the stated hypotheses?

Reviewer #1: Yes

3. Is the methodology feasible and described in sufficient detail to allow the work to be replicable?

Reviewer #1: Yes

4. Have the authors described where all data underlying the findings will be made available when the study is complete?

Reviewer #1: Yes

5. Is the manuscript presented in an intelligible fashion and written in standard English?

Reviewer #1: Yes

6. Review Comments to the Author

You may also provide optional suggestions and comments to authors that they might find helpful in planning their study.

Reviewer #1: Dear Authors,

When I reviewed your revised article titled "Probiotic Bacteria vs. Yeast for Gastrointestinal Diseases Treatment: Protocol for a Systematic Review and Meta-Analysis" that you applied as a study protocol, again, I believe that it is a protocol that can be beneficial to the field of study. I see the minor corrections that I suggested, done. Thanks for your attention.

7. PLOS authors have the option to publish the peer review history of their article (what does this mean? ). If published, this will include your full peer review and any attached files.

**Do you want your identity to be public for this peer review?** For information about this choice, including consent withdrawal, please see our Privacy Policy .

Reviewer #1: No

---

## [Editor Report · Acceptance letter]

PONE-D-24-50095R2

PLOS ONE

Dear Dr. Alshawsh,

I'm pleased to inform you that your manuscript has been deemed suitable for publication in PLOS ONE. Congratulations! Your manuscript is now being handed over to our production team.

Kind regards,

on behalf of

Professor Awatif Abid Al-Judaibi

Academic Editor

PLOS ONE